# *FAM111A* Is a Novel Molecular Marker for Oocyte Aging

**DOI:** 10.3390/biomedicines10020257

**Published:** 2022-01-25

**Authors:** Huixia Yang, Thomas Kolben, Mirjana Kessler, Sarah Meister, Corinna Paul, Julia van Dorp, Sibel Eren, Christina Kuhn, Martina Rahmeh, Cornelia Herbst, Sabine Gabriele Fink, Gabriele Weimer, Sven Mahner, Udo Jeschke, Viktoria von Schönfeldt

**Affiliations:** 1Department of Obstetrics and Gynecology, University Hospital, Ludwig-Maximilians-University, 81377 Munich, Germany; huixia.yang@med.uni-muenchen.de (H.Y.); thomas.kolben@med.uni-muenchen.de (T.K.); mirjana.kessler@med.uni-muenchen.de (M.K.); sarah.meister@med.uni-muenchen.de (S.M.); corinna.paul@med.uni-muenchen.de (C.P.); julia.van.dorp@hotmail.de (J.v.D.); sibel.eren@med.uni-muenchen.de (S.E.); christina.kuhn@uk-augsburg.de (C.K.); martina.rahmeh@med.uni-muenchen.de (M.R.); cornelia.herbst@med.uni-muenchen.de (C.H.); s.fink@med.uni-muenchen.de (S.G.F.); gabriele.weimer@med.uni-muenchen.de (G.W.); sven.mahner@med.uni-muenchen.de (S.M.); viktoria.schoenfeldt@med.uni-muenchen.de (V.v.S.); 2Department of Obstetrics and Gynecology, University Hospital Augsburg, 86156 Augsburg, Germany

**Keywords:** oocyte aging, cellular metabolism, DNA replication, histone modifications, cell cycle, WNT signaling pathway, *FAM111A*

## Abstract

Aging is the main cause of decline in oocyte quality, which can further trigger the failure of assisted reproductive technology (ART). Exploring age-related genes in oocytes is an important way to investigate the molecular mechanisms involved in oocyte aging. To provide novel insight into this field, we performed a pooled analysis of publicly available datasets, using the overlapping results of two statistical methods on two Gene Expression Omnibus (GEO) datasets. The methods utilized in the current study mainly include Spearman rank correlation, the Wilcoxon signed-rank test, *t*-tests, Venn diagrams, Gene Ontology (GO), Protein–Protein Interaction (PPI), Gene Set Enrichment Analysis (GSEA), Gene Set Variation Analysis (GSVA), and receiver operating characteristic (ROC) curve analysis. We identified hundreds of age-related genes across different gene expression datasets of in vitro maturation-metaphase II (IVM-MII) oocytes. Age-related genes in IVM-MII oocytes were involved in the biological processes of cellular metabolism, DNA replication, and histone modifications. Among these age-related genes, *FAM111A* expression presented a robust correlation with age, seen in the results of different statistical methods and different datasets. *FAM111A* is associated with the processes of chromosome segregation and cell cycle regulation. Thus, this enzyme is potentially an interesting novel marker for the aging of oocytes, and warrants further mechanistic study.

## 1. Introduction

Female fertility decreases progressively with age. It has been revealed that the fertility of a woman in her mid-30s is only half of the fertility potential of a woman in her mid-20s; and this is reduced to less than 5% for a woman in her 40s [1]. The decrease in fertility is reflected in the lower quantity and quality of oocytes. Age-related decline in oocyte quality is an important non-male cause of assisted reproductive technology (ART) failures, and there is no successful treatment for the age-related decline in oocyte quality [2]. Increased aneuploidy is an important cause of the decline in oocyte quality [3]. In addition, oocyte quality decline has also been associated with nuclear and mitochondrial DNA damage [4,5], meiotic spindle abnormalities, chromosomal misalignment, mitochondrial dysfunction [6], telomere shortening [7], and abnormal structure of the zona pellucida [8,9] and nucleolus [10].

To date, several investigations have explored the influence of maternal age on oocyte gene expression, using microarray and RNA-seq technology. Of these, most have been single-center studies on one or two types of oocytes, with limited sample size. Different analysis methods were used in each study, and the results of these studies remain controversial. Some studies reported that gene expression data for the human germinal vesicle (GV) oocytes [11,12], from women across age groups, remained stable; while others revealed that gene expression among GV oocytes [13], in vivo ovulation-metaphase Ⅱ (IVO-MII) oocytes [14,15,16,17], and in vitro maturation-metaphase II (IVM-MII) oocytes [12,13], were significantly influenced by maternal age. To obtain more generalizable results, multi-center studies with large sample sizes are required.

Given the above research background, in this study, we performed an integrative analysis of the available datasets. The data were derived from two original Gene Expression Omnibus (GEO) datasets, which provide gene expression profiles of oocytes from women of various ages. These two datasets were deposited, respectively, by Reyes et al. [12] and Llonch et al. [13]. In the study of Reyes et al. [12], age was treated as a categorical variable, and differential gene expression analysis between young and old groups was performed using the DESeq2 package. In the study of Llonch et al. [13], age was treated as a continuous variable, and Pearson correlation was applied to investigate the association between gene expression and age. In the current study, age was treated as a categorical variable, and the overlapping results of limma moderated *t*-tests and Spearman correlation were obtained. The primary aim of this study was to explore oocyte age-related genes, which were validated in a second dataset, and to investigate the potential biological functions affected by these age-related genes using functional enrichment analysis. We believe our results provide more reliable insight into understanding the molecular mechanisms underlying the age-related decline in oocyte quality, which may facilitate the development of targeted treatments designed to improve oocyte quality.

## 2. Materials and Methods

### 2.1. Data Download and Processing

Gene expression profiles of oocytes (Accession No. GSE95477 [12], GSE158802 [13], and GSE87201 [17]) were obtained from the NCBI GEO (http://www.ncbi.nlm.nih.gov/geo/, accessed on 9 August 2021) database. Data processing included quantile normalization and log_2_-transformation. All R packages were run in R software (version 4.1.0).

### 2.2. Age-Related Analysis

Next, as oocyte quality declines with age, we treated age as a categorical variable (using median age to stratify the oocytes), and examined the correlation between age and gene expression by performing Spearman rank correlation, using the R function ‘cor.test’. For each detected gene, the Spearman correlation value (*r*) and *p*-value were calculated. Genes with absolute value *r* > 0.3 and *p*-value < 0.05 were considered significantly correlated with age.

Differential gene expression analysis between young and old groups was performed using R package ‘limma’ (version 3.44.3) [18] and moderated *t*-tests. Genes with absolute fold change (FC) value > 2 and *p*-value < 0.05 were considered significant.

Then, Venn diagrams were generated using ‘jvenn’ [19] in order to visualize the overlapping results of Spearman rank correlation and *t*-tests, for IVM-MII oocytes from GSE95477 (referred to as Group I in the current study) and GSE158802 (referred to as Group II in the current study).

Based on the overlapping results presented in the Venn diagrams, the overlapping age-related genes in different IVM-MII oocytes were further validated for GV oocytes from GSE158802 (referred to as Group III in the current study) and IVO-MII oocytes from GSE87201 (referred to as Group IV in the current study), using the same methods and parameters as described above.

### 2.3. Gene Ontology (GO) Functional Enrichment Analysis

To gain an insight into the biological functions of the age-associated genes, Gene Ontology (GO) functional enrichment analysis was performed using package ‘ClusterProfiler’ (version 3.12.0) [20], with the overlapping age-related genes as the input gene list. The R packages ‘ggplot2’ (version 3.3.3) [21] and ‘GOplot’ (version 1.0.2) [22] were used to plot the results. GO categories with *p*-value < 0.05 were considered as significantly enriched.

### 2.4. The Capability of Hub Gene FAM111 Trypsin-like Peptidase A (FAM111A) for Distinguishing Young and Old Oocytes

The *FAM111 Trypsin-Like Peptidase A* (*FAM111A*) expression levels between young and old groups were compared using *t*-tests (for normally distributed expression data) or Wilcoxon signed-rank tests (for non-normally distributed expression data) in R package ‘ggpubr’ (version 0.4.0) [23].

Receiver operating characteristic (ROC) curves and the area under the curve (AUC) were obtained via R package ‘pROC’ (version 1.15.3) [24] and visualized using R package ‘ggplot2’ (version 3.3.3) [21]. The AUC had values generally ranging between 0.5 and 1. Typically, AUC results are categorized as ‘uninformative’ (AUC = 0.5), ‘less accurate’ (0.5 < AUC ≤ 0.7), ‘moderately accurate’ (0.7 < AUC ≤ 0.9), or ‘very accurate’ (0.9 < AUC < 1) [25].

### 2.5. Gene Set Enrichment Analysis (GSEA) and Gene Set Variation Analysis (GSVA)

To explore the association between *FAM111A* expression and biological processes/signaling pathways, Gene Set Enrichment Analysis (GSEA) and Gene Set Variation Analysis (GSVA) were carried out, with *FAM111A* as the input gene list. GSEA is a computational method for determining whether a given gene set exerts statistically significant and concordant differences between two cohorts/phenotypes [26]. GSEA analyses of IVM-MII oocytes from Group I (GSE95477) and Group II (GSE158802) were performed using ‘ClusterProfiler’ (version 3.12.0) [20] with default parameters. Terms with *p*-value < 0.05 were considered statistically significant. GSVA is a non-parametric and unsupervised algorithm that transforms genes in a sample matrix into pre-defined gene sets, without the need for knowing in advance the experiment design [27]. GSVA analyses of IVM-MII oocytes from Group II (GSE158802) were individually implemented via the R package ‘GSVA’ (version 1.30.0) [27] with default parameters. Terms with *p*-value < 0.05 and absolute value r > 0.3 were considered significant. All GO and KEGG related gene sets were obtained from the Molecular Signature Database (MsigDB, http://software.broadinstitute.org/gsea/msigdb/, accessed on 28 August 2021).

### 2.6. Protein–Protein Interaction (PPI) Network Based on FAM111A

Protein–Protein Interaction (PPI) networks were constructed for FAM111A and its interacting proteins using the GeneMANIA database [28]. GeneMANIA is a user-friendly and regularly updated web interface for predicting gene function, analyzing gene lists, and extending the given gene list with genes of similar function using available proteomics and genomics data [29]. It has been widely applied in the study of PPI relationships. The PPI networks were constructed in terms of co-expression, physical interaction, co-localization, genetic interaction, common pathways, shared protein domains, etc.

### 2.7. Correlation Analysis of FAM111A and Its Interacting Proteins

To explore the potential association between *FAM111A* and its interacting proteins, gene expression profiles of *FAM111A* and its interacting proteins were extracted from oocytes from Group I and Group II. R package ‘SVA’ (version 3.32.1) [30] was used to remove batch effects. The gene-gene correlation matrix between *FAM111A* and its interacting partners was generated using R package ‘corrplot’ (version 0.84) [31] with Spearman correlations. Genes with absolute value r > 0.3 and *p*-value < 0.05 were considered significantly correlated.

## 3. Results

A summary of the gene expression profiles included in the current analysis is shown in Figure 1A, and Figure 1B presents an overview of the analysis procedures of the study. Based on the median age of each group, the oocytes were stratified into young and old oocytes. The rationale for using the median age to stratify young and old oocytes is the same as that for female fertility, with a downward trend continuing with age. The major objective of our study is to explore the differences presented by oocytes that are already visible at the molecular level.

### 3.1. Impact of Age on the Gene Expression of Human Oocytes

Based on Spearman correlation, IVM-MII oocytes from Group I (GSE95477) and Group II (GSE158802) presented 1322 and 933 genes, respectively, showing altered expression with age (Appendix A). Based on *t*-tests, IVM-MII oocytes from Group I and Group II presented, respectively, 377 and 480 genes, with different expression levels between young and old oocytes (Appendix A).

We used Venn diagrams (Figure 2) to identify age-related genes with overlapping Spearman correlation and *t*-test results. There were 249 age-related genes with overlapping results in oocytes from Group I (Figure 2B), 397 age-related genes with overlapping results in oocytes from Group II (Figure 2C), and three age-related genes with overlapping results (*FAM111A*, *Isopentenyl-Diphosphate Delta Isomerase 1* (*IDI1*), and *Lymphocyte Antigen 6 Family Member K* (*LY6K*)) in all the analysis results (Figure 2D,E).

To test the validity of our findings (i.e., three overlapping age-related genes), we analyzed GV oocytes from Group III and IVO-MII oocytes from Group IV, and confirmed the correlation of *FAM111A* with age. The correlation coefficients of *FAM111A*, *IDI1*, and *LY6K* were, respectively, −0.32 (*p* = 0.039), −0.11 (*p* = 0.470), and 0.08 (*p* = 0.602), in Group III. The correlation coefficients of *FAM111A*, *IDI1*, and *LY6K* in Group Ⅳ were, respectively, 0.34 (*p* = 0.046), 0.05 (*p* = 0.796), and −0.09 (*p* = 0.605).

### 3.2. Cellular Metabolism, DNA Replication, and Histone Modifications Are the Biological Processes Enriched by Age-Related Genes in Two Datasets

GO enrichment analysis, based on the age-related genes with overlapping Spearman correlation and *t*-test results, revealed that cellular metabolism, DNA replication, and histone modifications were the biological processes significantly affected by age in both Group I (Figure 3A) and Group II (Figure 3B). GO enrichment analysis, based on the age-related genes with overlapping Spearman correlation results from Group I and Group II, also revealed that biological processes, such as the oxidation of fatty acids/lipids, fatty acid catabolic process, and histone modifications, were significantly influenced by age (Figure 4).

### 3.3. FAM111A Is the Hub Age-Related Gene in In Vitro Maturation-Metaphase II (IVM-MII) Oocytes and FAM111A Expression Is also Correlated with Age in Germinal Vesicle (GV) and In Vivo Ovulation-Metaphase II (IVO-MII) Oocytes

Based on *t*-test and Wilcoxon signed-rank test results, we found that *FAM111A* expression not only showed significant differences between young and old oocytes from Group I (Figure 5A) and Group II (Figure 5B), but also showed significant differences in GV oocytes from Group III (Figure 5C) and IVO-MII oocytes from Group Ⅳ (Figure 5D). Based on the ROC analysis, *FAM111A* showed good discrimination between young and old oocytes (Figure 5E–H).

### 3.4. Chromosome Segregation and Regulation of Cell Cycle Are the Main Pathways Regulated by FAM111A in IVM-MII Oocytes

Based on the overlapping results of the GSEA analysis of IVM-MII oocytes from Group I (Appendix A) and Group II (Appendix A), and GSVA analysis of IVM-MII oocytes from Group II (Appendix A), meiotic chromosome segregation, the regulation of the cell cycle, the positive regulation of the WNT signaling pathway, actin filament organization, and microtubule-associated complex, were terms significantly regulated by *FAM111A* (Table 1).

### 3.5. Predicted FAM111A-Associated Protein Includes Zinc Finger Protein 226 (ZNF226), Which Functions in Transcriptional Regulation

We used the GeneMANIA database to analyze the FAM111A-associated proteins (Figure 6A). These proteins were predicated based on a highly adaptive algorithm and hundreds of datasets, which were collected from ten publicly available databases, such as GEO, BioGRID, Interologous Interaction Database (I2D), and Pathway Commons [29]. The results reveal that FAM111A interacted with Proliferating Cell Nuclear Antigen (PCNA), Mediator Complex Subunit 18 (MED18), Zinc Finger Protein 226 (ZNF226), Nicastrin (NCSTN), and Kringle Containing Transmembrane Protein 2 (KREMEN2), etc. Further, Spearman correlation analysis of the FAM111A-associated proteins, based on the gene expression profiles of the oocytes, revealed a borderline correlation between *FAM111A* and *ZNF226* (correlation coefficient *r* = 0.32; *p* = 0.054) (Figure 6B).

## 4. Discussion

In this study, we explored the influence of age on oocytes at the gene expression level. To increase the robustness of our findings, we explored the age-related genes using two statistical methods; moreover, we attempted to identify overlapping results by combining two independent datasets. We found hundreds of genes with significantly altered expression levels between oocytes from young and old age groups; the biological processes influenced by these genes mainly included cellular metabolism, DNA replication, and histone modifications. Among these age-related genes, *FAM111A* expression displayed robust correlations with age in the different statistical methods and different datasets. To our knowledge, this is the first report of a potential gene marker for human oocyte aging. We found that *FAM111A* may be involved in the biological processes associated with chromosome segregation and the regulation of the cell cycle. FAM111A protein could potentially interact with regulators of transcription (i.e., ZNF226) and transcription factors (i.e., PCNA).

We identified 249 overlapping age-related genes in oocytes from Group I, and 397 overlapping age-related genes in oocytes from Group II. Based on GO enrichment analysis, we found these two gene lists were both enriched in biological processes associated with cellular metabolism, DNA replication, and histone modifications. The overlapping genes, from Spearman rank correlation, in oocytes from Group I and Group II were also enriched in the terms of the oxidation of fatty acids/lipids, fatty acid catabolic process, and histone modifications. Similarly, Duncan et al. found that age-associated gene signatures in mouse oocytes were involved in the alteration of protein metabolism [10]. Another study, based on liquid and gas chromatography coupled to mass spectrometry, also found that total free fatty acids were decreased in old equine oocytes compared with young oocytes. Furthermore, through the quantification of aerobic and anaerobic metabolism, these researchers found that the metabolic activity of equine oocytes was impaired by maternal aging [32]. Fatty acids are vital substrates for early reproductive events (e.g., oocyte maturation [33,34] and embryo implantation [35]). Upon demand, the metabolism of lipids provides a good source of energy during oocyte maturation. For example, during pig oocyte maturation, fatty acids in lipid droplets (LDs) provide adenosine triphosphate (ATP) via β-oxidation, serving as an energy source for oocytes [36]. The lipid modulators have been investigated to improve the developmental competence of oocytes and embryos in animal models [37]. Modifications to LD morphology and lipid metabolism during the period of oocyte maturation have been revealed to possibly interfere with monospermic fertilization and embryo development in the pig model [38,39]. It has been revealed that the degradation of LDs tends to be via the lipolysis of triacylglycerols to fatty acyl-CoA by lipases at the surface of the LDs, which is often in preparation for mitochondrial metabolism [40].

Corroborating our results, the alteration of histone methylation [41,42] and histone acetylation [43,44,45] has been reported in old mammalian oocytes. Altered histone modifications usually correlated with changes in chromatin configuration and gene transcription activity [46], which may affect oocyte fertilization and embryo development [44]. For example, decreased histone 4 lysine 20 tri-methylation (H4K20me3) results in chromosomal segregation defects in cell culture [47]. The inhibitor of histone deacetylase during IVM contributes to the reduced developmental potential of bovine oocytes [48]. Notably, epigenetic changes have been regarded as a hallmark of aging, besides telomere shortening [49]. However, studies of age-related epigenetic changes in oocytes are still finite, and, therefore, further studies on this topic are highly awaited.

We identified *FAM111A* as the most robust gene that presented altered gene expression during aging. The overlapping results of GSEA and GSVA analysis based on *FAM111A* revealed that chromosome segregation and regulation of the cell cycle were the main pathways regulated by *FAM111A* in IVM-MII oocytes from both datasets. Other studies on human oocytes [13,50] have come to similar conclusions, that transcripts associated with chromosome segregation and cell cycle regulation are altered during oocyte aging. Fine et al. revealed that *FAM111A* expression is cell-cycle-dependent in cell-cycle-synchronized T98G cells [51]. FAM111A is a chymotrypsin-like serine protease, and displays autocleavage activity in vivo [52]. It has been revealed that FAM111A is important for DNA replication [52,53] and plays a role in viral restriction [51,54]. Furthermore, it is hypothesized that FAM111A might engage in DNA binding [52], as it localizes to replication forks [55]. Nevertheless, until now, the functions of FAM111A remain largely unclarified and require further investigation.

We also found that the WNT signaling pathway was positively regulated by *FAM111A* in IVM-MII oocytes, and that *FAM111A* expression was decreased in aging IVM-MII oocytes. Similarly, a recent study also found that the WNT signaling was downregulated in low-quality rescue IVM (rIVM) oocytes, and revealed that down-regulation of the GATA-1/CREB1/WNT signaling axis might result in the poor maturation performance of rIVM oocytes [56]. This might indicate the existence of a potential link between *FAM111A* expression and WNT signaling, which needs to be further investigated. The WNT signaling pathway is a highly conserved pathway. It plays a major role in cell and tissue maintenance, polarity, and differentiation [57], as well as in maintaining female germ cell survival [58]; furthermore, it is frequently dysregulated in human cancers [59]. The canonical WNT pathway is initiated by WNT proteins [60], which have been revealed as being responsible for regulating the cell cycle and deciding cell fate [61].

Furthermore, based on our PPI network results, we found that FAM111A protein may have interactions with ZNF226 and PCNA. The Spearman correlation analysis based on the gene expression profiles from Group I and Group II also revealed a borderline positive correlation between *FAM111A* expression and *ZNF226* expression. PCNA is a transcription factor that promotes DNA polymerase delta binding to DNA [62], and plays an essential role in DNA replication, DNA repair [63], and cell cycle regulation [62]. It has also been reported that ZNF226 plays a vital role in transcriptional regulation [64]. Our finding reveals a potential link between transcriptional regulation and oocyte aging.

Though the same statistical methods were used in different datasets, there was still variation in age-related genes between the different datasets, with relatively few overlapping results. This could be due to differences in sequencing methods and coverage of the sequencing platforms, as well as differing age ranges and sources of the study participants. In addition, for the in vitro maturation of oocytes, no unified IVM protocols are established across different in vitro fertilization (IVF) centers [65]. However, differences in methods of follicular priming, time of oocyte retrieval, choices of IVM medium and supplementation, and duration of in vitro culture, may also contribute towards differential gene expression in IVM oocytes. Another limitation is that, in our differential gene expression analysis, *p*-values (and not adjusted *p*-values) less than 0.05 were considered statistically significant. This lack of compensation may increase the frequency of false-positive results. However, we extracted the overlapping results with different analytical methods and datasets, which to some extent could reduce variability and provide relatively reliable results.

## 5. Conclusions

In summary, based on two independent datasets of IVM-MII oocytes, we identified hundreds of genes that were influenced by age, and that might be involved in the biological processes associated with cellular metabolism, DNA replication, and histone modifications. Among these age-related genes, *FAM111A* may act as a biomarker for oocyte aging, and deserves further investigation. Our findings could advance current knowledge of the molecular mechanisms involved in oocyte aging, and potentially facilitate the development of targeted treatments designed to improve oocyte quality.

## Figures and Tables

**Figure 1 biomedicines-10-00257-f001:**
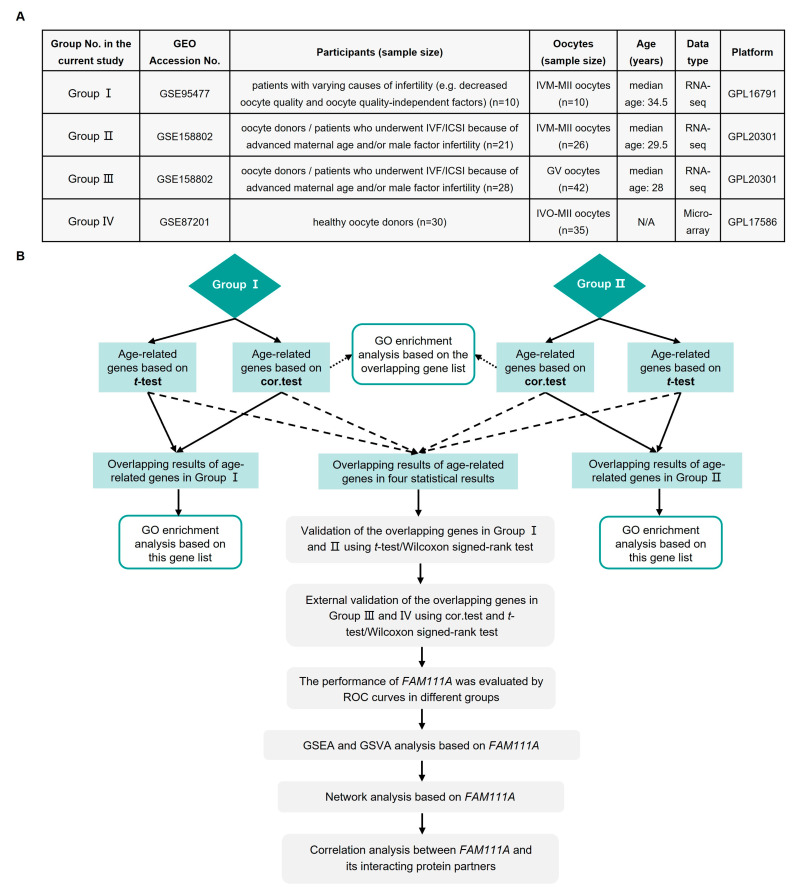
Summary of the present study. (**A**) Groups included in the analysis of this study. (**B**) Flowchart of this study. GEO, Gene Expression Omnibus; IVF/ICSI, in vitro fertilization/intracytoplasmic sperm injection; IVM-MII, in vitro maturation-metaphase II; GV, germinal vesicle; IVO-MII, in vivo ovulation-metaphase II; N/A, not available; cor.test, Spearman rank correlation; GO, Gene Ontology; ROC, receiver operating characteristic; GSEA, Gene Set Enrichment Analysis; GSVA, Gene Set Variation Analysis; *FAM111A*, *FAM111 Trypsin-Like Peptidase A*.

**Figure 2 biomedicines-10-00257-f002:**
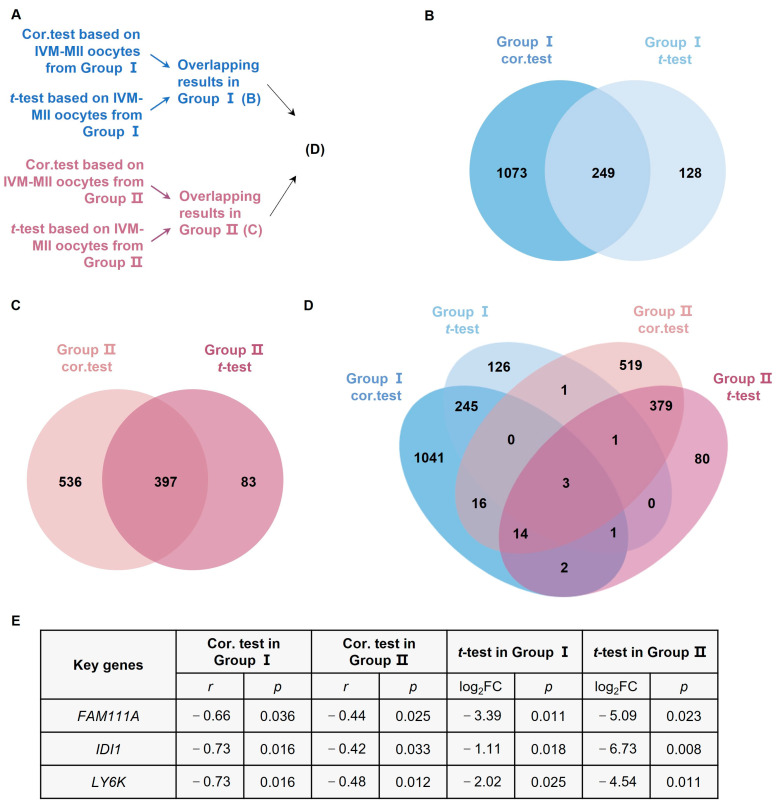
Analysis of age-related genes. (**A**) Flowchart of the age-related analysis. (**B**) Overlapping results of cor.test and *t*-test for IVM-MII oocytes from Group I. (**C**) Overlapping results of cor.test and *t*-test for IVM-MII oocytes from Group II. (**D**) Overlapping results of age-related genes in Group I and Group II. (**E**) Based on the cor.test and *t*-test, *FAM111A*, *IDI1*, and *LY6K* are the statistically significant age-related genes in both Group I and Group II. *IDI1*, *Isopentenyl-Diphosphate Delta Isomerase 1*; *LY6K*, *Lymphocyte Antigen 6 Family Member K*. FC, fold change.

**Figure 3 biomedicines-10-00257-f003:**
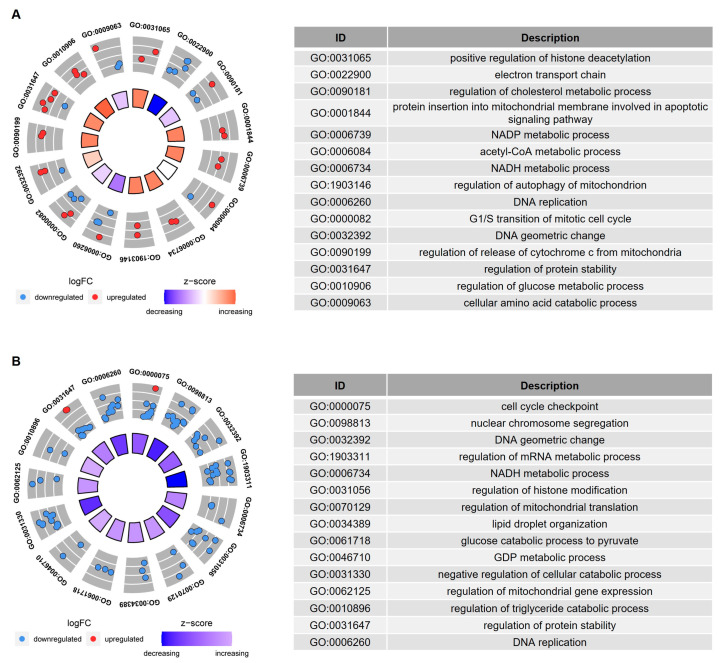
GO enrichment analysis of the age-related genes in IVM-MII oocytes from (**A**) Group I and (**B**) Group II. The age-related genes in this analysis were obtained from the overlapping cor.test and *t*-test results. The inner circle in the left figure indicates the Z-score value. The outer ring represents the significantly enriched terms, and the most upregulated and downregulated genes. The blue dot indicates the downregulated gene and the red dot indicates the upregulated gene. The right figure is the description of the enriched GO terms.

**Figure 4 biomedicines-10-00257-f004:**
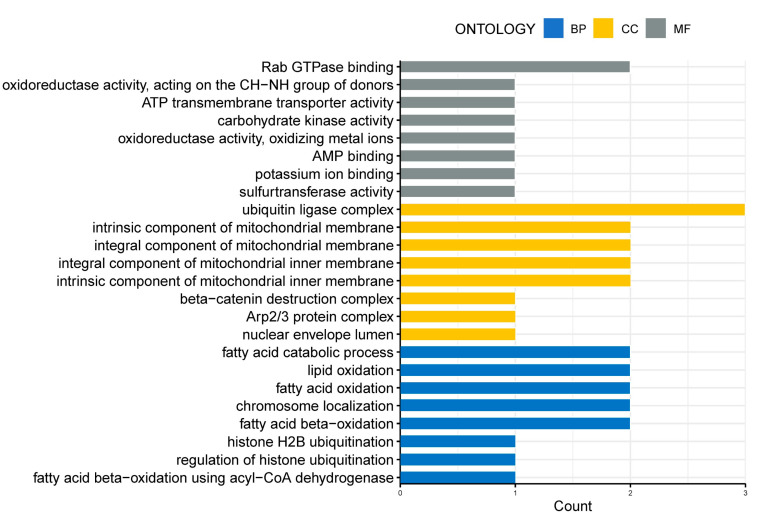
GO enrichment analysis of the overlapping age-related genes in IVM-MII oocytes from Group I and Group II. The age-related genes in this analysis were obtained from the overlapping results of cor.test in Group I and Group II. The *x*-axis represents the counts of the genes enriched in GO terms. The *y*-axis represents the significantly enriched terms in BP, CC, and MF. BP, biological process; CC, cellular component; MF, molecular function.

**Figure 5 biomedicines-10-00257-f005:**
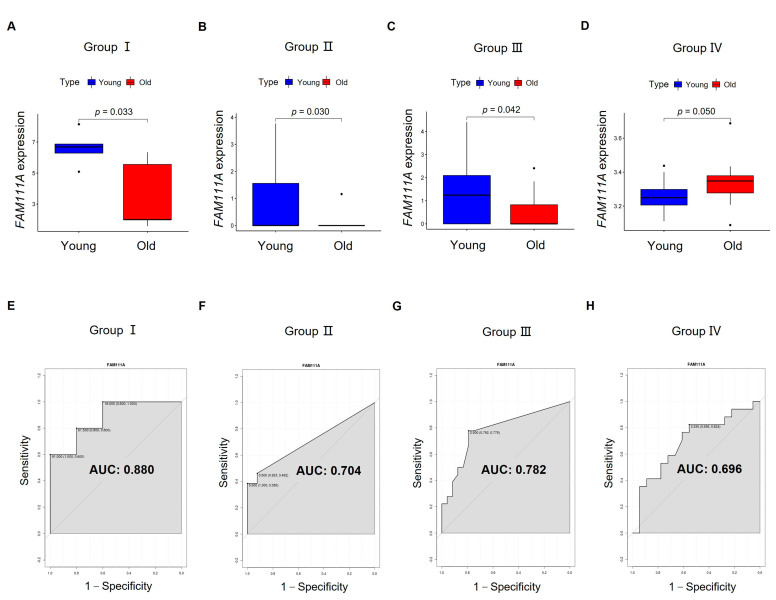
The capability of *FAM111A* for distinguishing young and old oocytes. The expression of *FAM111A* in the oocytes from young and old groups is visualized in (**A**) Group I and (**B**) Group II, and further validated in (**C**) Group III and (**D**) Group IV. The *x*-axis represents young and old oocytes. The *y*-axis represents *FAM111A* expression. *p*-values were calculated using a *t*-test or Wilcoxon signed-rank test. ROC curves are visualized in (**E**) Group I, (**F**) Group II, (**G**) Group III, and (**H**) Group Ⅳ. The *x*-axis of the ROC graph represents 1—specificity, and the *y*-axis represents sensitivity. AUC, area under the curve.

**Figure 6 biomedicines-10-00257-f006:**
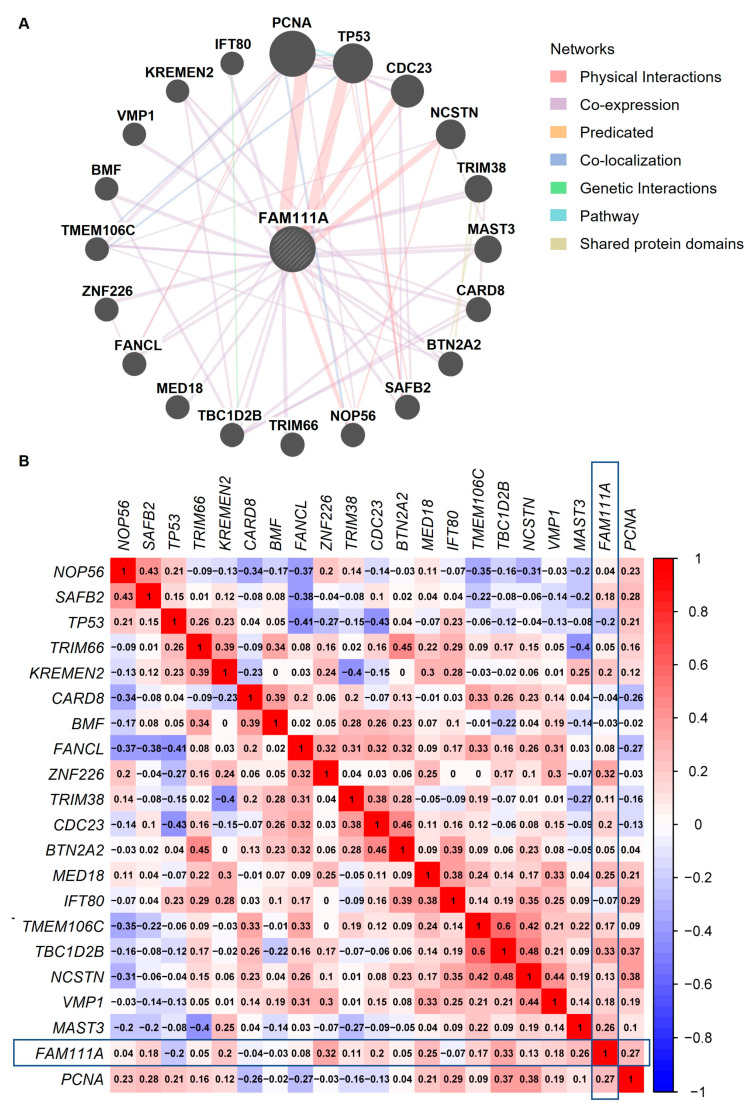
The PPI network and correlation analysis between FAM111A and its interacting protein partners. (**A**) The PPI network of FAM111A and its interacting protein partners. (**B**) Spearman gene-gene correlation matrix between *FAM111A* and its interacting partners. In the matrix, red and blue represent positive and negative correlations, respectively. The number in each square indicates the Spearman correlation coefficient (*r*-value). PPI, Protein–Protein Interaction.

**Table 1 biomedicines-10-00257-t001:** The terms overlapping in the results of GSEA and GSVA based on IVM-MII oocytes.

IVM Terms	GSEA of Group I		GSEA of Group II		GSVA of Group II	
	NES	*p*	NES	*p*	*r*	*p*
GO_MEIOTIC_CHROMOSOME_SEGREGATION	1.65	0.003	1.48	0.004	0.43	0.030
GO_CONDENSED_NUCLEAR_CHROMOSOME_CENTROMERIC_REGION	1.53	0.038	1.71	0.003	0.61	<0.001
GO_CHROMOSOME_LOCALIZATION	1.66	0.004	1.40	0.018	0.44	0.024
GO_REGULATION_OF_CELL_CYCLE_G2_M_PHASE_TRANSITION	1.41	0.010	1.44	<0.001	0.47	0.015
GO_ACTIN_FILAMENT_ORGANIZATION	1.36	0.009	1.36	<0.001	0.51	0.008
GO_POSITIVE_REGULATION_OF_WNT_SIGNALING_PATHWAY	1.39	0.019	1.45	<0.001	0.43	0.028
GO_MICROTUBULE_ASSOCIATED_COMPLEX	1.40	0.019	1.43	0.002	0.53	0.006
GO_PROTEASOME_ACCESSORY_COMPLEX	1.75	0.006	1.52	0.021	0.43	0.027
GO_EXODEOXYRIBONUCLEASE_ACTIVITY	1.71	0.010	1.46	0.038	0.49	0.011

NES > 0, the terms were positively regulated by *FAM111A*; *r* > 0, the terms were positively correlated with *FAM111A*. NES, normalized enrichment score.

## Data Availability

Publicly available datasets were analyzed in this study. These data can be found here: GEO: https://www.ncbi.nlm.nih.gov/geo/ (accessed on 15 November 2021).

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
