# Peer review of "FAM111A Is a Novel Molecular Marker for Oocyte Aging"

_biomedicines, 2022, doi:10.3390/biomedicines10020257_

Round 1
Reviewer 1 Report
In this study, the authors performed a meta-analysis of the transcriptome profile of GV and MII oocytes with respect to age using the publicly available data sets (Accession No. GSE95477 and GSE158802. According to the authors' report, including FAM111A, several age-related genes that are involved in cellular metabolism, DNA replication, and histone modifications were identified. Moreover, based on the results obtained from correlation and differential expression analysis, the authors claimed that FAM111A expression is a novel molecular marker for oocyte aging although they reported several genes. Although the study seems to be relevant in the field, there are several issues that were expected to be covered by this study.
- This manuscript did not describe the differences and similarities of findings obtained from the current study and the results reported by the original data owners (Reyes et al., 2017 and Llonch et al. 2021)
- The title of the manuscript is very much tempting. Indeed, based on correlation, differential expression, and ROC curves analysis, the FAM111A gene expression was found to be different between young and old oocyte groups. Nevertheless, the authors should also consider the limitation that was forwarded by the original data owners(GSE95477, Reyes et al., 2017) which stated “Although the approach has the benefit of identifying intrinsic differences between samples, it may not be completely representative of in vivo matured oocyte”. Thus to claim this title, the authors need to perform further independent in vivo studies.
- Why the authors did perform a correlation analysis for group1 and group II oocytes separately instead of pooling the data together and performing one correlation analysis?
- Based on the median age of each group, oocytes were stratified as young and old groups but the authors did not mention the median age that is specific to the young and old groups.
- The authors performed a differential gene expression analysis between the young and old group while this type of analysis was already done by the authors who submitted the GSE9547 and GSE158802 data.
- As indicated in supplemental tables 3 and 4, the adjusted p-value of all transcripts reported as significantly differentially expressed is 1 although the raw p-value is < 0.05. The adjusted p-value is similar to the FDR. Therefore, it is not clear why the authors reported those genes as DEG although the adjusted p-value is 1. The authors need to clarify the statistical power of those differentially expressed genes.
Author Response
Reviewer 1
In this study, the authors performed a meta-analysis of the transcriptome profile of GV and MII oocytes with respect to age using the publicly available data sets (Accession No. GSE95477 and GSE158802. According to the authors' report, including FAM111A, several age-related genes that are involved in cellular metabolism, DNA replication, and histone modifications were identified. Moreover, based on the results obtained from correlation and differential expression analysis, the authors claimed that FAM111A expression is a novel molecular marker for oocyte aging although they reported several genes. Although the study seems to be relevant in the field, there are several issues that were expected to be covered by this study.
- This manuscript did not describe the differences and similarities of findings obtained from the current study and the results reported by the original data owners (Reyes et al., 2017 and Llonch et al. 2021).
Answer: Thank you very much for your comments, the common denominator is that we all explored the age-related genes in human oocytes. While the analytical methods differed among different studies. In the study of Reyes et al., age was treated as a categorical variable, differential gene expression analysis between young and old groups was performed using DESeq2 package. In the study of Llonch et al., age was treated as a continuous variable, Pearson correlation analysis was applied to investigate the associations between gene expression levels and age. While, in the current study, age was treated as a categorical variable, both the differential gene expression analysis (using the limma package) and the correlation analysis (using Spearman correlation analysis) were performed to explore the age-related genes. The specific methodology employed here is different from the previous two studies. Besides, the findings of current study are based on the overlapping results of different analytical methods (i.e., limma differential expression analysis and Spearman correlation analysis) and the overlapping results of different datasets (i.e., from Reyes et al., 2017 and Llonch et al. 2021). The description of the similarities and differences has been added in the manuscript (please see lines 60—68).
- The title of the manuscript is very much tempting. Indeed, based on correlation, differential expression, and ROC curves analysis, the FAM111A gene expression was found to be different between young and old oocyte groups. Nevertheless, the authors should also consider the limitation that was forwarded by the original data owners(GSE95477, Reyes et al., 2017) which stated “Although the approach has the benefit of identifying intrinsic differences between samples, it may not be completely representative of in vivo matured oocyte”. Thus to claim this title, the authors need to perform further independent in vivo studies.
Answer: Thank you for your insightful comments, we have performed independent validation in in vivo matured oocytes (IVO-Mâ…¡) using the public dataset (GEO Accession No. GSE87201). (â…°) Based on Spearman correlation test, we found FAM111A was significantly correlated with age (Spearman correlation coefficient is 0.34 (p = 0.046)). (â…±) Based on t-test, we found the expression of FAM111A between young and old groups was perceivable statistical significance (p = 0.050). (â…²) Based on ROC curve analysis, we found the area under the curve is 0.696. The independent validation of in vivo matured oocytes has been added in the manuscript (please see line 77, lines 100,101, line 183, lines 186,187, lines 217,218, lines 222,224).
To further explore the role of FAM111A in oocyte aging, we also performed time-series analysis to illustrate the dynamic changes of gene expression following prolonged in vitro culture time (<18h→24h→30h). As we known, the prolonged in vitro culture time may also contribute oocyte aging. In detail, after removing batch effect, we find the co-expression genes among IVM-MII oocytes from GSE158539 [1] (in vitro culture time < 18 hours), GSE95477 [2] (in vitro culture time 24 hours), GSE158802 [3] (in vitro culture time 30 hours). Secondly, based on the co-expression genes, one-way analysis of variance (ANOVA) was performed between the three groups. Genes with p-value < 0.05 were regarded as differentially expressed genes and further subjected to time-series analysis via R package ‘MFuzz’ (version 2.40) [4]. ‘MFuzz’ performs soft clustering of genes based on their expression values using the fuzzy c-means algorithm.
We found that the expression of 3,574 genes was significantly altered during prolonged IVM time. Six gene clusters with different expression dynamics were identified (as shown in the following figure). To be noted, FAM111A was the member of cluster 2 (membership: 0.59), showing a significantly altered expression during prolonged IVM time.
- Why the authors did perform a correlation analysis for group1 and group II oocytes separately instead of pooling the data together and performing one correlation analysis?
Answer: Thank you for your comments, considering the egg sources, follicular priming methods, IVM medium and the supplements, and IVM time varied among different datasets, and these differences could not be totally eliminated by ‘remove batch effects function’ of the R package…To avoid the introduction of batch effects, we choose to analyze each dataset separately and then take intersections of the results.
This is also the reason that we did not put the results of time-series analysis into the current manuscript. Even though, the mRNA expression of FAM111A is significantly decreased during prolonged IVM time in the time-series analysis.
- Based on the median age of each group, oocytes were stratified as young and old groups but the authors did not mention the median age that is specific to the young and old groups.
Answer: It was mentioned in the Figure 1A.
- The authors performed a differential gene expression analysis between the young and old group while this type of analysis was already done by the authors who submitted the GSE9547 and GSE158802 data.
Answer: Thank you for your comments! We carried out a different correlation analysis (using Spearman correlation test) combined with a different differentially expressed gene (DEG) analysis (using limma package). The results obtained from DESeq2 package (used by Reyes et al.) and from limma package are different. And the results from Pearson (used by Llonch et al.) and Spearman correlation analysis are different. Besides, our findings are based on the overlapping results of different analytical methods and the overlapping results of different datasets, which can increase the credibility of the present study.
With the rapid development of bioinformatics, more and more new bioinformatic methods/packages/tools have been developed for bioinformatic analysis.. It has been revealed that limma package has a better ability to find the accurate DEGs than DESeq2 package [5,6].
- As indicated in supplemental tables 3 and 4, the adjusted p-value of all transcripts reported as significantly differentially expressed is 1 although the raw p-value is < 0.05. The adjusted p-value is similar to the FDR. Therefore, it is not clear why the authors reported those genes as DEG although the adjusted p-value is 1. The authors need to clarify the statistical power of those differentially expressed genes.
Answer: Thank you for your careful review! Indeed, this p-value is adjusted by Benjamini–Hochberg method, it is false discovery rate (FDR) exactly. We have checked three independent public human oocyte datasets (GSE158539 [1], GSE95477 [2], and GSE158802 [3]), we found, when the oocytes were divided into young and old groups using the median age, the FDRs in the DEG analysis are all equal or close to 1 (the results are available in the supplementary material). It seems that FDR=1 is a frequent phenomenon in the analysis of DEGs. Two reasons may account for this phenomenon. (1) The sample size is too small to produce a significant FDR; (2) The inherent properties of the data model itself could also contribute to this phenomenon. In the current study, most of the oocytes are derived from the healthy oocyte donors or the women underwent IVF/ICSI because of advanced maternal age and/or male factor infertility, the control and test groups were divided based on median age. It is reasonable that the biomolecular differences between these two groups were moderate, and less pronounced than the normal tissue versus cancer tissue groups. Indeed, it is necessary to require the FDR of DEGs being less than 0.05 when we carry out the DEG analysis on the data from normal and cancer tissue groups. But for the oocyte age-related analysis, based on the public oocyte datasets, the FDRs of the DEG analysis are all equal or close to 1. This phenomenon may be due to the inherent properties of this kind of project. But, in spite of lower statistical power of differentially expressed genes in oocyte aging than those in oncology, it is still meaningful to explore the DEGs in oocyte aging, as it could advance knowledge on the molecular mechanisms involved in oocyte aging and potentially facilitate the development of targeted treatments to improve oocyte quality.
In the recently published peer-reviewed papers, the DEGs are also defined with p value <0.05 [7,8].
Thank you very much for your comments. Wish you a happy and successful 2022!
Reference:
- Lee, A.W.T.; Ng, J.K.W.; Liao, J.; Luk, A.C.; Suen, A.H.C.; Chan, T.T.H.; Cheung, M.Y.; Chu, H.T.; Tang, N.L.S.; Zhao, M.P.; et al. Single-cell RNA sequencing identifies molecular targets associated with poor in vitro maturation performance of oocytes collected from ovarian stimulation. Reprod. 2021, 36, 1907–1921, doi:10.1093/humrep/deab100.
- Reyes, J.M.; Silva, E.; Chitwood, J.L.; Schoolcraft, W.B.; Krisher, R.L.; Ross, P.J. Differing molecular response of young and advanced maternal age human oocytes to IVM. Reprod. 2017, 32, 2199–2208, doi:10.1093/humrep/dex284.
- Llonch, S.; Barragán, M.; Nieto, P.; Mallol, A.; Elosua-Bayes, M.; Lorden, P.; Ruiz, S.; Zambelli, F.; Heyn, H.; Vassena, R.; et al. Single human oocyte transcriptome analysis reveals distinct maturation stage-dependent pathways impacted by age. Aging Cell 2021, 20, doi:10.1111/acel.13360.
- Kumar, L.; Futschik, M.E. Mfuzz: A software package for soft clustering of microarray data. Bioinformation 2007, 2, 5–7, doi:10.6026/97320630002005.
- Tong, Y. The comparison of limma and DESeq2 in gene analysis. E3S Web Conf. 2021, 271, 3058, doi:10.1051/e3sconf/202127103058.
- Merino, G.A.; Conesa, A.; Fernández, E.A. A benchmarking of workflows for detecting differential splicing and differential expression at isoform level in human RNA-seq studies. Bioinform. 2019, 20, 471–481, doi:10.1093/bib/bbx122.
- Sun, G.; Chen, J.; Liang, J.; Yin, X.; Zhang, M.; Yao, J.; He, N.; Armstrong, C.M.; Zheng, L.; Zhang, X.; et al. Integrated exome and RNA sequencing of TFE3-translocation renal cell carcinoma. Commun. 2021, 12, doi:10.1038/s41467-021-25618-z.
- Böhme, J.; Martinez, N.; Li, S.; Lee, A.; Marzuki, M.; Tizazu, A.M.; Ackart, D.; Frenkel, J.H.; Todd, A.; Lachmandas, E.; et al. Metformin enhances anti-mycobacterial responses by educating CD8+ T-cell immunometabolic circuits. Commun. 2020, 11, doi:10.1038/s41467-020-19095-z.

Reviewer 2 Report
This is a very interesting high quality manuscript.
The influence of age on human oocyte was studied at the gene expression level, overlapping results by combining two independent datasets. The analysis procedures of the study showed interesting results that were refined by GO enrichment analysis. Although hundreds of genes were influenced by age and might be involved in the biological processes associated with cellular metabolism, DNA replication, and histone modifications,at the end, the overlapping results of the 2 IVM-MII groups and the cor.test and DEG analysis resulted in 3 age-related genes. The validation of the findings was confirmed analyzed GV oocytes, confirming the correlation of FAM111A with age.
These results are very interesting, considering that in most studies hundreds of genes are differentially expressed in oocytes according to the age of the animal. This large number of genes makes understanding difficult and prevents the establishment of a good molecular marker of oocyte aging. In this sense, the identification of FAM111A as the most robust gene that presented altered gene expression during aging was an advance towards the establishment of a cellular aging biomarker. The identification of this gene is particularly relevant, since the main pathways regulated by FAM111A in IVM-MII oocytes were chromosome segregation and regulation of cell cycle.
Author Response
Reviewer 2
This is a very interesting high quality manuscript.
The influence of age on human oocyte was studied at the gene expression level, overlapping results by combining two independent datasets. The analysis procedures of the study showed interesting results that were refined by GO enrichment analysis. Although hundreds of genes were influenced by age and might be involved in the biological processes associated with cellular metabolism, DNA replication, and histone modifications,at the end, the overlapping results of the 2 IVM-MII groups and the cor.test and DEG analysis resulted in 3 age-related genes. The validation of the findings was confirmed analyzed GV oocytes, confirming the correlation of FAM111A with age.
These results are very interesting, considering that in most studies hundreds of genes are differentially expressed in oocytes according to the age of the animal. This large number of genes makes understanding difficult and prevents the establishment of a good molecular marker of oocyte aging. In this sense, the identification of FAM111A as the most robust gene that presented altered gene expression during aging was an advance towards the establishment of a cellular aging biomarker. The identification of this gene is particularly relevant, since the main pathways regulated by FAM111A in IVM-MII oocytes were chromosome segregation and regulation of cell cycle.
Answer: Many thanks for your appreciation of our work. Wish you have a most happy and prosperous New Year!

Round 2
Reviewer 1 Report
The manuscript is now improved and the majority of my queries are addressed. However, considering genes with the adjusted p-value =1 as differentially expressed is very ambiguous. The authors have described the sample size and the inherent properties of the data to be the causes for the higher adjusted p-value. Nevertheless, statistical analysis is done to omit all those non-biological variations between samples and in the end to find statically sound differences (if any) between the experimental groups.
Author Response
Reviewer 1
The manuscript is now improved and the majority of my queries are addressed. However, considering genes with the adjusted p-value =1 as differentially expressed is very ambiguous. The authors have described the sample size and the inherent properties of the data to be the causes for the higher adjusted p-value. Nevertheless, statistical analysis is done to omit all those non-biological variations between samples and in the end to find statically sound differences (if any) between the experimental groups.
Answer: Thank you very much for your comment! We fully agreed with you. Indeed, only the “adjusted p-value < 0.05” has sufficient statistical strength/significance to guarantee the difference is not caused by individual variations. In this study, to reduce the influence of individual variance, all data were quantile normalized and log2 transformated prior to analysis. Meanwhile, to increase the statistical power of this study, we took the overlapping results of two statistical methods (i.e., Spearman correlation analysis and Limma t-test) in two datasets.
Considering your comment, we have made a more cautious conclusion (please see lines 380 and 381, in red font), and have discussed this shortcoming in the limitation section (please see lines 370—373, in red font).
Meanwhile, we have canceled the usage of “differentially expressed gene” or “DEG” throughout the whole manuscript. In detail, we have modified (1) “DEG analysis” to “t-test” (please see the Method and Result sections in red font, Figure 1, and Figure 2), (2) “Genes with absolute fold change (FC) value>2 and p-value < 0.05 were regarded as DEG” to “Genes with absolute fold change (FC) value>2 and p-value < 0.05 were considered significant” (please see line 89, in red font), (3) “DEG” to “genes with different expression levels” (please see lines 171 and 172, in red font).
Thank you again for your precious time and efforts in reviewing our manuscript!
